# Serum Uric Acid Levels and Risk of Rapid Decline of Estimated Glomerular Filtration Rate in Patients with Type 2 Diabetes: Findings from a 5-Year Prospective Cohort Study

**DOI:** 10.3390/healthcare9101341

**Published:** 2021-10-09

**Authors:** Hoa Tuyet Le, Tung Thanh Le, Nguyet Minh Thi Tran, Thuy Thanh Thi Nguyen, Ni Chanh Su Minh, Quyen Thi Le, Tuyet Anh Thi Tram, Thang Duc Tran, Tung Xuan Doan, Mai Huynh Thi Duong, Truc Thanh Thai

**Affiliations:** 1Faculty of Internal Medicine, School of Medicine, Vietnam National University, Ho Chi Minh City 700000, Vietnam; hoalt@pnt.edu.vn; 2Faculty of Internal Medicine, Pham Ngoc Thach School of Medicine, Ho Chi Minh City 700000, Vietnam; 3Outpatient Clinic, District 10 Hospital, Ho Chi Minh City 700000, Vietnam; thanhtung1064@yahoo.com.vn (T.T.L.); bsminhnguyetq10@yahoo.com.vn (N.M.T.T.); thuyphuc02@yahoo.com (T.T.T.N.); bsni0810@gmail.com (N.C.S.M.); quyenley16@gmail.com (Q.T.L.); tramthianhtuyet@gmail.com (T.A.T.T.); thangtranrgkg2507@gmail.com (T.D.T.); bsdaotung@gmail.com (T.X.D.); 4Faculty of Public Health, University of Medicine and Pharmacy at Ho Chi Minh City, Ho Chi Minh City 700000, Vietnam; huynhmaiyhdp14@gmail.com

**Keywords:** type 2 diabetes, serum uric acid, rapid decline, estimated glomerular filtration rate

## Abstract

This study investigated the association between serum uric acid (SUA) levels with rapid decline of the estimated glomerular filtration rate (eGFR) in type 2 diabetes (T2 DM) patients. A prospective cohort study was conducted in a community-based hospital in Vietnam. We followed 405 T2DM patients with normal kidney function for five years. Rapid progression of kidney function was defined as an average annual decrease of eGFR of at least 4 mL/min/1.73 m^2^ and was found in 16.0% of patients. Patients in the SUA high tertile ( ≥6 mg/dL) had higher BMI (*p* = 0.004), lower HbA1c (*p* = 0.001), lower eGFR (*p* < 0.001) and higher rate of hypertension than low and middle tertile. After adjusting for age and sex, rapid progression of renal function was significantly associated with SUA level (OR = 1.22, 95% CI 1.02–1.45, *p* = 0.026). This association was marginally significant when more covariates were included in the model (OR = 1.20, 95% CI 0.99–1.46, *p* = 0.065). However, the association between tertiles of SUA and rapid decline of eGFR was not statistically significant. This study demonstrates neither a strong significant association between SUA and rapid decline of eGFR nor evidence to refuse the role of SUA levels in the increased risk of renal function decline in in T2DM patients.

## 1. Introduction

Almost half of patients with type 2 diabetes mellitus (T2DM) have chronic kidney disease (CKD) [1]. Nephropathy related to T2DM remains a leading cause of end-stage renal disease, resulting in an increased burden on individuals and healthcare systems. The estimated glomerular filtration rate (eGFR) for evaluating changes in kidney function is the most widely-used parameter in clinical practice. However, the natural progression of eGFR is complex and heterogenous in type 2 diabetes [2]. Therefore, an understanding of risk factors associated with the progression of CKD in patients with T2DM would be useful to preserving kidney function.

Many risk factors for rapid decline in renal function have been determined, including hyperglycemia [3], hypertension [3,4], and macroalbuminuria [5]. Previous studies have documented that elevated serum uric acid (SUA) levels are positively associated with the development of diabetic nephropathy [6,7]. In Vietnam, a recent cross-sectional study reported that patients with low eGFR had higher albumin excretion and evidence of more proliferative retinopathy and hyperuricemia than those with normal eGFR [8]. Prior epidemiologic studies have suggested that higher SUA levels are significantly associated with renal disease development [9,10]. However, it remains unclear as to whether higher SUA is independently associated with rapid decline of renal function leading to end-stage renal disease [11,12].

The present study aimed to investigate the association of SUA levels and risk stratification for rapid decline in eGFR over a five-year follow-up in Vietnamese patients with T2DM. Findings from this study can help to design prevention strategies and to optimize treatments for patients with T2DM.

## 2. Materials and Methods

### 2.1. Setting and Participants

Data presented in this paper were from a prospective observational cohort project where patients with T2DM were evaluated the progression rate of eGFR. In brief, during 2014–2016, we enrolled 467 patients aged ≥35 and <80 years-old who were diagnosed with T2DM for at least five years and received clinical examination in an outpatient setting at the district hospital in Ho Chi Minh City, Vietnam. Details about setting and participants were described elsewhere [8]. Patients with polycystic kidney, kidney tumor, only one kidney or urinary obstruction, advanced concurrent disease (including stroke, heart failure, unstable chest pain, hepatic failure, gout arthritis, pulmonary tuberculosis, thyroid dysfunction), pregnant and lactating women were excluded. In this analysis, 405 patients who had normal kidney functions with eGFR ≥ 60 mL/min/1.73 m^2^ at baseline and were followed up for at least three years were included (Figure 1). Of these, no patient received allopurinol or febuxostat due to the absence of gout arthritis symptoms. All 405 patients were followed until October 2019, with a median follow-up of 5 years (IQR 4–5).

### 2.2. Clinical and Biochemical Measurement

All patients received a comprehensive clinical examination including documentation of current and past medical history, history of hypertension medications and anti-hyperglycemia drugs. In addition, baseline HbA1c, lipid profile, SUA, serum creatinine, and the urine albumin-creatinine ratio were collected. 

Blood samples were captured in steady-state patients. The serum creatinine concentrations were measured by Jaffe assay and were ascertained by biannual examinations [13]. The eGFR was calculated using the CKD-EPI equation [14]. The mean of the two stable eGFR in the same year was considered as the eGFR of the individual in the respective year. Albuminuria was determined by the immune-turbidity method using polyclonal anti-albumin antibodies [15]. The mean urine albumin-to-creatinine ratio (ACR) from two separate early morning specimens was recorded. SUA was measured by couple enzyme reaction. The HbA1c levels were determined by a high-performance liquid chromatography method using an automated analyzer. The value for HbA1c is estimated as a National Glycohemoglobin Standardization Program equivalent value (%) as recommended by the American Diabetes Association (ADA) [16]. The serum total cholesterol and triglyceride were measured by enzyme method. 

### 2.3. Diagnosis Criteria

Type 2 diabetes mellitus status was defined by 2014 ADA definition or current history of diabetes mellitus [16]. Overweight/obesity was based on WHO criteria for the Asian population [17]. Hypertension was defined as systolic blood pressure (SBP) ≥ 140 mmHg and/or diastolic blood pressure (DBP) ≥ 90 mmHg or being on antihypertensive medication. According to the 2014 ADA Guideline, hypercholesterolemia or hypertriglyceride was defined as total cholesterol ≥ 200 mg/dL or triglycerides ≥ 150 mg/dL [16]. The definition for decreased eGFR was in line with the current KDIGO 2013 guideline [13]. Normal eGFR was determined as eGFR ≥ 60 mL/min/1.73 m^2^. 

### 2.4. Outcome Variable

Rapid decline in eGFR was based on measures of creatinine level after at least three years. Rates of change were calculated by the following equation: (eGFR at last follow-up—eGFR _at baseline_)/Number of follow-up years. Since the mean decrease rate of eGFR in this study was 1.4 (SD 3.9) mL/min/1.73 m^2^ per year, rapid eGFR decline was defined as an average annual decrease of 4 mL/min/1.73 m^2^ which was approximately three times higher than the average rate. 

### 2.5. Statistical Analysis

All variables were compared between participants with and without rapid decline in kidney function. Normality of data distribution was checked using quantile-quantile plots, histograms and Shapiro-Wilk tests [18]. Student’s *t*-tests, one-way ANOVA tests, Mann-Whitney U tests, Kruskal Wallis tests and the Chi-squared tests were used for between-group comparisons of the continuous and categorical variables when appropriate. To evaluate the effect of SUA on the rapid decline of eGFR, three forms of SUA were used, including continuous values, the epidemiological termination for hyperuricemia and tertiles. Hyperuricemia was confirmed if SUA was >7 mg/dL in males and >5.7 mg/dL in females. The subjects were stratified into tertiles, including low-tertile: ≤4.86 mg/dL; middle-tertile: 4.87 - <6.00 mg/dL and high-tertile: ≥6.00 mg/dL). 

Multivariate logistic regression models were used to evaluate the association between SUA levels and rapid decline in eGFR. Variables selected for inclusion in final adjusted models were based on whether they were statistically different between the two groups. Moreover, factors that have been shown to be associated with either SUA levels or the rapid decline of eGFR were adjusted and retained in the models (including age, sex, BMI, diabetes duration, HbA1c, systolic blood pressure, cholesterol, triglyceride, ACR, baseline eGFR). The results were shown as odds ratios and its 95% confidence interval. In all statistical tests, a *p* value of less than 0.05 was considered statistically significant.

### 2.6. Ethics

This project was approved by the Ethics Committee of Pham Ngoc Thach School of Medicine and the District 10 hospital. All participants read the participation information sheet and signed the written informed consent.

## 3. Results

### 3.1. Participants’ Characteristics 

Among 432 patients with normal eGFR at baseline, we excluded 27 patients from the analysis because they visited the clinic only twice during the five-year follow-up. The demographic and clinical characteristics at baseline of the 405 patients are shown in Table 1. The mean age of the study participants was 61.2 (SD = 7.8) years. Most were female (68.1%) and non-smokers (82.2%). Over 60% of patients were overweight and obese. The mean diabetes duration was 8.8 (SD = 4.6) years. The mean HbA1c was 7.9% (SD = 1.6%). At least 40% of patients had increased ACR, and only 6.9% of total patients had severe albuminuria.

The baseline biochemical and laboratory characteristics of the patients are also shown in Table 1. Higher SUA levels were found in patients with lower eGFR (*p* < 0.001), lower HbA1c (*p* < 0.001), hypertension (*p* = 0.048), and higher BMI (*p* = 0.004). There were 121 patients (29.9%) with hyperuricemia.

### 3.2. Serum Acid Uric and Rapid Decline in eGFR 

Of the 405 patients, 65 (16.0%) had rapid decline of eGFR during the follow-up period. The baseline biochemical and clinical characteristics of the patients in the two groups (rapid decline or non-rapid decline) are presented in Table 2. No difference was found between the two groups on blood pressure status, baseline HbA1c, eGFR, HUA, and albuminuria. Patients with rapid decline of eGFR had hypercholesterolemia, higher triglyceride, higher SUA levels and higher frequency of taking diuretic drugs than those with non-rapid decline of eGFR.

The magnitude of annual change in eGFR stratified by SUA tertiles and HUA status during follow-up is depicted in Figure 2. The decrease of eGFR was greater in the high-tertile and in the HUA group. The percentage of decrease in eGFR was 7.1% among patients in the SUA high-tertile, and 6.2% among patients in the low-tertile (*p* = 0.327). The percentage of decrease in eGFR were 7.7% and 5.3% among patients with HUA and with normal SUA level, respectively (*p* = 0.099).

The baseline SUA levels were significantly associated with rapid decline of eGFR (OR = 1.19, 95% CI 1.01–1.41, *p* = 0.044) in the univariate analysis (Table 3). After adjusting for age and sex, the SUA levels remained significantly associated with the rapid decline of eGFR (OR = 1.22 95% CI 1.02–1.45, *p* = 0.026). After adjusting for other covariates, including, BMI, diabetes duration, HbA1c, systolic blood pressure, cholesterol, triglyceride, ACR, baseline eGFR, and diuretic use, the association between SUA and rapid kidney function decline was marginally significant (OR = 1.20, 95% CI 0.99–1.46, *p* = 0.065). However, analysis of the association of the SUA tertiles with rapid decline of eGFR in Table 4 indicated that both the high and middle SUA tertile were not statistically associated with rapid decline in eGFR.

## 4. Discussion

The main findings of our study are as follows: (1) there was no clear evidence of the association between SUA levels and rapidly declining kidney function; (2) while there was a moderate number of patients with HUA (almost 30% of subjects), all of them were asymptomatic (i.e., no gout arthritis symptoms). The percentage of HUA in the current study seems similar to that of previously reported studies [19,20]. There were 31% of men and 19.6% of women with HUA reported in another study at a provincial hospital [20]. Similar findings were also found in other countries. A study in Japan revealed that 32% of men and 15% of women with diabetes had HUA [21]. In Italy, a hospital-based study revealed 28.9% of 842 patients had hyperuricemia [22].

We observed a prevalence of rapid decline in eGFR of 16.0% in patients with normal kidney function during five-year follow-up. The rate is alarming given that in a community-based setting most subjects had been defined as having normal kidney function and few demonstrated proteinuria. This finding may suggest rapid precipitation to end-stage renal disease. Our current study finds the rate is much higher than has been previously reported. Zoppini et al. observed that from a cohort of 1682 participants with normal renal function, 263 (15.6%) proceeded to develop rapid decline in eGFR (defined as >4% per year after 10 years) [4]. Another study documenting a wider follow-up period of 0 to 22 years reported 29.3% of their cohort showed creatinine doubling [23]. 

In this prospective cohort study, we did not observe a clear association between SUA levels and rapidly declining kidney function (annual loss ≥ 4 mL/min/1.73 m^2^), despite a previously reported association in a cross-sectional study [8]. The association between baseline SUA level and subsequent progressive decrease in eGFR was only significant with adjustments for sex and age. However, this association was not statistically significant when more pathologic covariates were adjusted. It is possible that the small sample size in our study led to non-significant findings in the presence of many covariates. In a meta-analysis, significant findings were reported only when studies included larger sample sizes (≥1000 participants) [24]. Chonchol et al. reported a moderate association between SUA and progression in kidney function (for SUA > 6 mg/dL, OR = 1.47 95% CI 1.04–2.07) in 5808 participants and 16.2% were T2DM patients [25]. A large, single center, 5-year retrospective cohort study including 8078 individuals with eGFR ≥ 60 mL/min/1.73 m^2^ revealed that 1 mg/dL increase in baseline SUA was related to 1.27 times greater in odds of rapid eGFR decline. Particularly, 1 mg/dL increase in SUA levels over 5-years was associated with 3.77-fold risk of rapid eGFR decline (OR = 3.77, 3.35–4.26) [26]. Although the current findings from this study were unable to fully elucidate the role of SUA in rapid decline of kidney function, future investigation is still warranted in studying SUA levels in T2DM with concomitant risk factor such as hypertension.

Although the findings demonstrated higher baseline SUA levels the greater the decrease in eGFR at follow-up, there remained no significant association with rapid decline, contrary to the findings reported in Chonchol’s study [25]. Chang reported that SUA levels were independently associated with CKD progression when the value exceeded 6.3 mg/dL [27]. Recently, in 360 T2DM Japanese patients, the odds of eGFR decline ( ≥3 mL/min/1.73 m^2^ per year) was higher in the higher tertiles (SUA > 5.86 mg/dL) compared to the low tertile (SUA < 5.86 mg/dL) [28]. The longitudinal data from a Japanese diabetes registry indicated a significant association between the SUA ≥ 8 mg/dL and rapid progression in kidney function (OR = 3.98, 95% CI 1.02–15.51) [29]. Of note, HUA (defined as SUA > 7 mg/dL in male and >5.6 mg/dL in female) was an independent risk factor not only for the development of incident chronic kidney disease [29,30], but also for the significantly decreased change in the eGFR [21]. 

The present study has several limitations worthy of mention. First, the major limitation is a small sample size that did not allow for a full exploration of covariate influence in the outcome. However, limited by small sample size, our study demonstrates the feasibility in a small community hospital setting. Future studies including larger sample sizes are necessary to better define and understand associations and their clinical relevance. Secondly, this is an observational study carried out in a single district hospital and thus the results cannot be generalized to patients in other settings. Finally, the error of eGFR when utilizing the CKD-EPI equation demonstrated a wide dispersion with an average of 30% [31]. This might lead to large variations in declining eGFR.

## 5. Conclusions

Early recognition of progressive loss of renal function warrants additional study of the underlying mechanisms responsible for renal failure. While progression and decline in renal function is multifactorial, this current study demonstrates neither a strong significant association between SUA and rapid decline of eGFR nor evidence to refute the role played by SUA levels in increasing the risk of renal function decline in T2DM patients. Future studies should be focused to further investigate covariates influencing renal function decline in T2DM patients.

## Figures and Tables

**Figure 1 healthcare-09-01341-f001:**
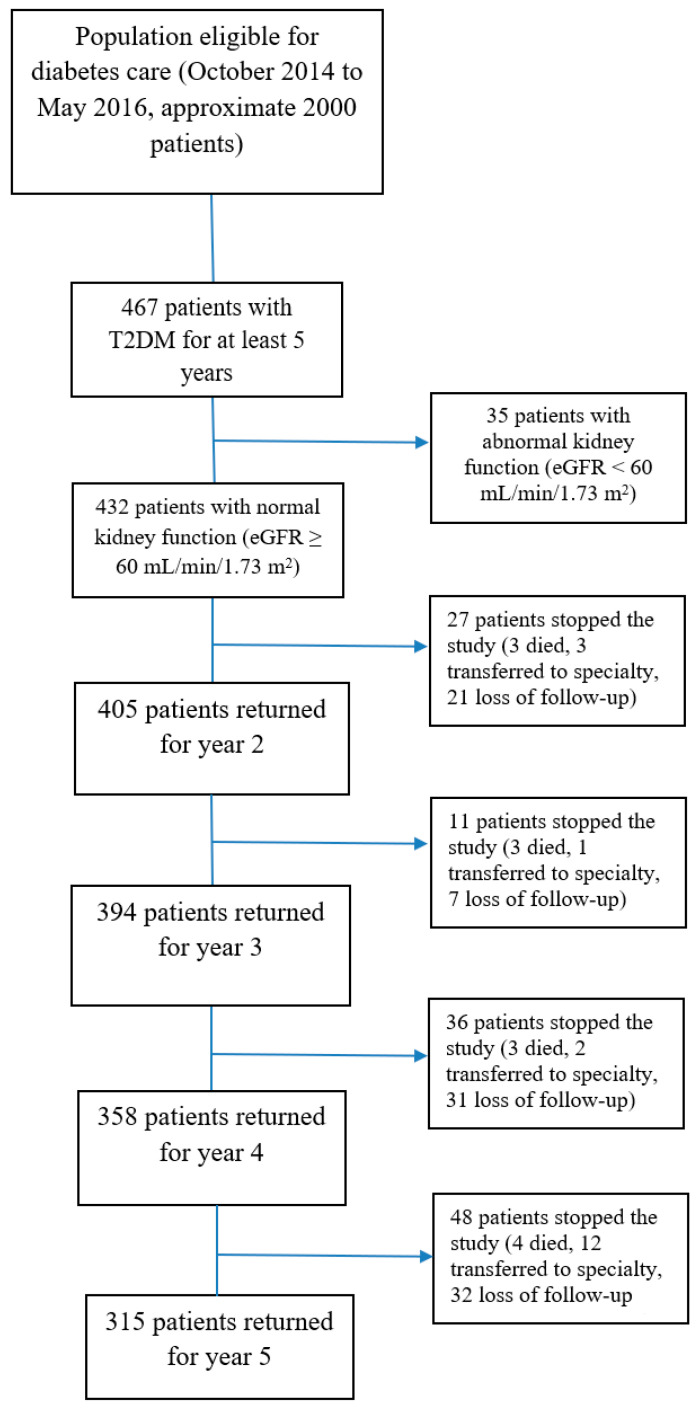
Study Flow Diagram. “Transferred to specialty”: the patients were transferred to nephrology, cardiology or neurology depending on the clinical severity.

**Figure 2 healthcare-09-01341-f002:**
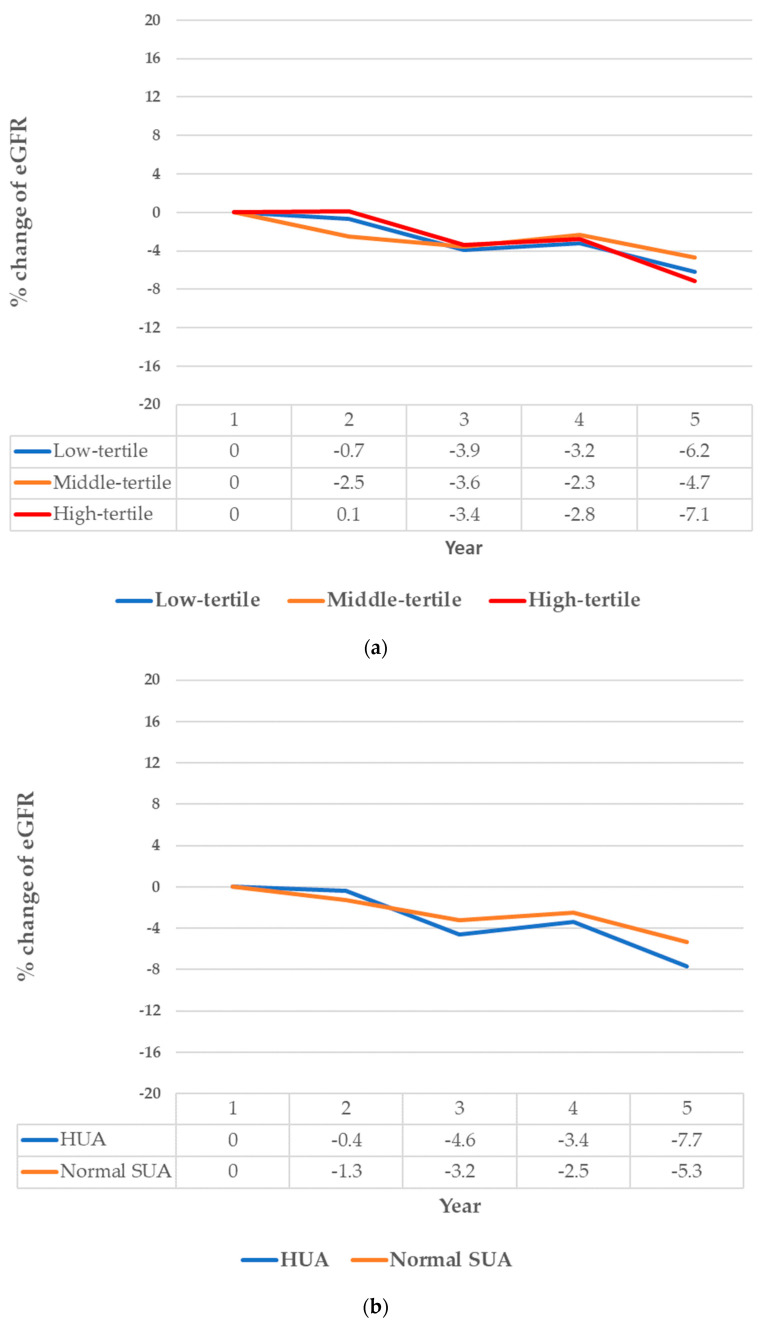
The decline of eGFR in five-year follow-up, stratified by tertile of serum uric acid (SUA), *p* = 0.327 (**a**) and hyperuricemia, *p* = 0.099 (HUA) (**b**).

**Table 1 healthcare-09-01341-t001:** Participant baseline characteristics for each of the serum uric acid tertiles.

Participant Characteristics at Baseline	All (*n* = 405)	Low-Tertile (*n* = 136)	Middle-Tertile (*n* = 136)	High-Tertile (*n* = 133)	*p* Value
Age, years [Mean (SD)]	61.2 (7.8)	60.1 (8.6)	61.9 (7.3)	61.7 (7.3)	0.130 ^a^
Sex, *n* (%)					
Male	129 (31.9)	36 (26.5)	42 (30.9)	51 (38.3)	0.108 ^c^
Female	276 (68.1)	100 (73.5)	94 (69.1)	82 (61.7)	
BMI, kg/m^2^ [Mean (SD)]	24.3 (3.4)	23.7 (3.1)	23.9 (2.8)	25.3 (4.0)	0.004 ^a^
Overweight/obesity, *n* (%)	253 (62.5)	81 (59.6)	84 (61.8)	88 (66.2)	0.523 ^c^
Smoking, *n* (%)	72 (17.8)	21 (15.4)	19 (14.0)	32 (24.1)	0.066 ^c^
Duration of diabetes, years [Mean (SD)]	8.8 (4.6)	8.5 (4.3)	8.6 (4.1)	9.1 (5.4)	0.763 ^a^
HbA1c (%) [Mean (SD)]	7.9 (1.6)	8.2 (1.4)	8.0 (1.8)	7.5 (1.4)	<0.001 ^a^
Type of treatment, *n* (%)					
Insulin use	89 (22.0)	36 (26.5)	28 (20.7)	25 (18.8)	0.286 ^c^
OAD	315 (78.0)	100 (73.5)	107 (79.3)	108 (81.2)	
Serum uric acid, mg/dL [Mean (SD)]	5.6 (1.5)	4.1 (0.6)	5.4 (0.3)	7.2 (1.1)	<0.001 ^a^
Hyperuricemia, *n* (%)	121 (29.9)	0 (0)	14 (10.3)	107 (80.5)	<0.001 ^c^
Creatinine, mg/dL [Mean (SD)]	0.8 (0.1)	0.7 (0.1)	0.8 (0.1)	0.8 (0.1)	<0.001 ^a^
eGFR (mL/min/1,73 m^2^) [Mean (SD)]	88.3 (12.5)	91.5 (12.4)	88.2 (12.7)	85.2 (11.7)	<0.001 ^a^
≥ 90 (G1), *n* (%)	195 (48.1)	74 (54.4)	74 (54.4)	47 (35.3)	0.001 ^c^
60–89 (G2)	210 (51.9)	62 (45.6)	62 (45.6)	86 (64.7)	
ACR, mg/g [Median (IQR)]	20.0 (11.7–54.5)	21.2 (12.2–64.4)	21.6 (12.6–60.6)	18.5 (10.4–48.6)	0.270 ^b^
Hypertension, *n* (%)	342 (84.4)	107 (78.7)	116 (85.3)	119 (89.5)	0.048 ^c^
Systolic BP, mmHg [Mean (SD)]	124.8 (11.5)	123.2 (11.0)	125.4 (12.0)	125.7 (11.5)	0.138 ^a^
Diastolic BP [Mean (SD)]	75.2 (8.0)	74.9 (7.9)	75.1 (7.8)	75.7 (8.4)	0.683 ^a^
ACEI/ARB use, *n* (%) (*n*= 345)	332 (97.1)	104 (97.2)	111 (95.7)	117 (98.3)	0.478 ^c^
Diuretic use, *n* (%)	28 (6.9)	6 (4.4)	10 (7.4)	12 (9.0)	0.319 ^c^
Total cholesterol, mg/dL [Mean (SD)]	177.3 (41.7)	181.7 (40.4)	175.4 (41.9)	174.8 (42.7)	0.319 ^a^
Triglyceride, mg/dL [Mean (SD)]	206.2 (110.5)	205.1 (104.4)	202.0 (109.2)	211.6 (118.3)	0.768 ^a^
Previous CVD, *n* (%)	146 (36.0)	43 (31.6)	56 (41.2)	47 (35.3)	0.254 ^c^

ACR, albumin/creatinine urine; BP, blood pressure; CVD, cardiovascular disease; OAD oral anti--hyperglycemic drug; SUA, serum uric acid. Values are reported as mean (SD), median (IQR) or as *n* (%), where indicated. ^a^ One-way ANOVA test; ^b^ Kruskal Wallis test; ^c^ Chi-squared test.

**Table 2 healthcare-09-01341-t002:** Participant baseline characteristics for each of the decline groups.

Characteristics	Rapid Decline eGFR (*n* = 65, 16.0%)	Non-Rapid Decline eGFR (*n* = 340, 84.0%)	*p*	OR (95% CI)
Age, years [Mean (SD)]	60.9 (7.4)	61.3 (7.9)	0.675 ^a^	0.99 (0.96–1.03)
Sex, *n* (%)				
Male	18 (27.7)	111 (32.6)	0.432 ^c^	0.79 (0.44–1.42)
Female	47 (72.3)	229 (67.4)		
BMI, kg/m^2^ [Mean (SD)]	24.5 (3.5)	24.3 (3.4)	0.658 ^a^	1.02 (0.94–1.10)
Duration of diabetes, years [Mean (SD)]	9.1 (4.4)	8.7 (4.7)	0.574 ^a^	1.02 (0.96–1.07)
HbA1c (%)[Mean (SD)]	8.1 (1.8)	7.9 (1.5)	0.248 ^a^	1.10 (0.94–1.29)
Diuretic use	9 (13.8)	19 (5.6)	0.028 ^c^	2.72 (1.17–6.30)
Type of treatment, *n* (%)				
Insulin use, *n* (%)	17 (26.2)	72 (21.2)	0.381 ^c^	1.31 (0.71–2.42)
OAD *n* (%)	48 (73.8)	267 (78.8)		
SUA, mg/dL [Mean (SD)]	5.9 (1.6)	5.5 (1.5)	0.042 ^a^	1.19 (1.01–1.41)
HUA, *n* (%)	22 (33.8)	99 (29.1)	0.445 ^c^	1.25 (0.71–2.19)
SUA tertiles, *n* (%)				
Low-tertile	16 (24.6)	120 (35.3)	0.228 ^c^	1
Middle-tertile	26 (40.0)	110 (32.4)		1.77 (0.90–3.48)
High-tertile	23 (35.4)	110 (32.4)		1.57 (0.79–3.12)
Creatinine, mg/dL [Mean (SD)]	0.8 (0.1)	0.8 (0.1)	0.852 ^a^	0.84 (0.13–5.40)
eGFR mL/min/1,73 m^2^ [Mean (SD)]	88.3 (11.6)	88.3 (12.7)	0.998 ^a^	1.00 (0.98–1.02)
ACR, mg/g [Median (IQR)]	24.7 (13.6–78.3)	19.9 (11.6–53.1)	0.188 ^b^	1.00 (1.00–1.00)
Hypertension, *n* (%)	56 (86.2)	286 (84.1)	0.678 ^c^	1.17 (0.55–2.52)
Systolic BP, mmHg [Mean (SD)]	126.5 (13.8)	124.4 (11.0)	0.193 ^a^	1.02 (0.99–1.04)
Diastolic BP	76.1 (8.1)	75.1 (8.0)	0.354 ^a^	1.02 (0.98–1.05)
ACEI/ARB use (*n* = 345), *n* (%)	54 (96.4)	278 (97.2)	0.671 ^c^	0.78 (0.16–3.76)
Total cholesterol mg/dL [Mean (SD)]	184.8 (48.9)	175.9 (40.1)	0.115 ^a^	1.01 (1.00–1.01)
Hypercholesterol- -emia, *n* (%)	27 (41.5)	86 (25.3)	0.007 ^c^	2.10 (1.21–3.64)
Triglyceride, mg/dL [Mean (SD)]	246.1 (139.9)	198.6 (102.5)	0.011 ^a^	1.00 (1.00–1.01)
Hypertriglyceride *n* (%)	47 (72.3)	214 (62.9)	0.148 ^c^	1.54 (0.86–2.76)

ACR, albumin/creatinine urine; BP, blood pressure; HUA, hyperuricemia; OAD oral anti-hyperglycemic drug; SUA, serum uric acid. Values are reported as mean (SD), median (IQR) or as *n* (%), where indicated. ^a^ Student’s *t*-test; ^b^ Mann-Whitney test; ^c^ Chi-squared test.

**Table 3 healthcare-09-01341-t003:** Association between serum uric acid concentration and rapid progression of renal function.

SUA	OR	95% CI	*p*
Crude	1.19	1.01–1.41	0.044
Model 1	1.22	1.02–1.45	0.026
Model 2	1.18	0.99–1.41	0.061
Model 3	1.20	0.99–1.46	0.065

Model 1: adjusted for age and sex. Model 2: adjusted for age, sex, cholesterol, triglyceride, and diuretic use. Model 3: adjusted for age, sex, BMI, diabetes duration, HbA1c, systolic blood pressure, cholesterol, triglyceride, ACR, baseline eGFR, and diuretic use.

**Table 4 healthcare-09-01341-t004:** Multivariate logistic regression on the relationship between the sua tertiles and rapid decline in eGFR.

Tertiles	OR	95% CI	*p*
Crude			
Low-tertile	1		
Middle-tertile	1.77	0.90–3.48	0.096
High-tertile	1.57	0.79–3.12	0.200
Model 1			
Low-tertile	1		
Middle-tertile	1.77	0.90–3.48	0.096
High-tertile	1.57	0.79–3.12	0.200
Model 2			
Low-tertile	1		
Middle-tertile	1.90	0.95–3.83	0.071
High-tertile	1.61	0.79–3.29	0.193
Model 3			
Low-tertile	1		
Middle-tertile	1.88	0.93–3.81	0.079
High-tertile	1.57	0.73–3.37	0.251

Model 1: adjusted for age and sex. Model 2: adjusted for age, sex, cholesterol, triglyceride, and diuretic use. Model 2: adjusted for age, sex, BMI, diabetes duration, HbA1c, systolic blood pressure, cholesterol, triglyceride, ACR, baseline eGFR, and diuretic use.

## Data Availability

The data presented in this study are available on request from the corresponding author. The data are not publicly available due to restrictions.

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
