# Peer review of "Serum Uric Acid Levels and Risk of Rapid Decline of Estimated Glomerular Filtration Rate in Patients with Type 2 Diabetes: Findings from a 5-Year Prospective Cohort Study"

_healthcare, 2021, doi:10.3390/healthcare9101341_

Round 1

Reviewer 1 Report

The study is interesting, and the manuscript well-written.

I have the following comments:

  1. In figure 1, the “transferred to specialty” has to be clarified better.
  2. The authors use the epidemiological termination for hyperuricemia in the text (> 7 mg/dL for males and > 5.6 mg /dL for females) instead the pathophysiological termination of > 7 mg/dL (the point of urate precipitation) for both sexes. Then they analyzed urate levels in both sexes and treated it as a continuous variable with the upper tertile group having urate > 6 mg/dL. In my opinion, the analysis is correct. However, to avoid confusion, the authors should explain the above with clarity in the revised manuscript.
  3. For comparison of means among tertiles, one-way ANOVA is more proper than the t-test.
  4. An exclusion criterion was the receiving of anti-hyperuricemic medications. This may be a selection bias since many patients with type 2 DM and severe hyperuricemia were excluded. This limitation should be noted.
  5. The authors should also note whether the patients started anti-hyperuricemic therapy during the 5-year follow-up. The latter may be a significant co-founder since xanthine oxidase plays an important role in ROS production under high glucose or shear stress conditions (Eleftheriadis et al., Int Urol Nephrol, doi: 10.1007/s11255-017-1733-5 / McNally et al., Am J Physiol Heart Circ Physiol, doi: 10.1152/ajpheart.00515.2003), and serum urate may signify its activity.
  6. The authors concluded that high serum urate does not contribute to rapid GFR decline in type 2 DM patients. Reading the manuscript, I have two concerns about this conclusion. First, after adjustment for many factors, the result almost reached statistical significance (OR 1.2, 95% CI 0.99-1.46). This should be noted in the revised manuscript. Second, the adjustment included factors that were not necessarily higher in the patients with high urate levels.

Reviewer 2 Report

Le et al. have carried out a clinical cohort study with which they try to assess whether there is a quantitative relationship between serum uric acid levels and the rapid deterioration of kidney function (measured in terms of glomerular filtration rate) in patients with type diabetes mellitus. 2. After carrying out the statistical methodology, they verify that it is not possible to fully demonstrate, but not to reject, such a relationship between these two parameters in their study population (from Vietnam).
Although it is an interesting study and could be used in the future to define new diagnostic systems for kidney failure in this type of population, several aspects need to be improved before its final publication:

Major points:

  • The introduction is excessively short. It is necessary to better define and expand the problem so that a reader unfamiliar with the subject can understand the objective of the work.
  • In point 2.2. some sample analysis techniques are cited. However, they are neither lightly described nor are any references included where they are described. This must be included.
  • Table 1: Does the p-value presented correspond to the comparison between which groups? It is not possible to really know which of the groups are different from each other. Furthermore, in the case that this p-value refers to all groups, this p-value could not be obtained with the method described in the Material and Methods section (since a Student's t test does not allow comparison between three or more groups, only between two). All this aspect must be corrected.

Minor points: 

  • Introduction, line 16: what does "normal kidney disease" mean?
  • Figure 1: Text is missing in all rectangles on the right.
  • Point 2.2. Line 62: Do you mean "blood or plasma samples"? because "creatinine samples" is not correct.
  • Point 2.2. Line 63: I think the correct term is "Jaffe".
  • Point 2.3. Line 74: The ADA abbreviation is not described in the text.
  • Point 2.5. Line 91: Was the normality of the data checked before applying the Student's test? With what test was it done?
  • Discussion. Line 172: "while there was a moderate 172 number of patients with HUA (almost 30% of subjects), some were asymptomatic"...What does it mean to be asymptomatic? What symptoms are you referring to?

Round 2

Reviewer 1 Report

The authors clarified all issues. 

Reviewer 2 Report

All changes have been successfully made.